# Learning Optimal Lattice Vector Quantizers for End-to-end Neural Image Compression

**Xi Zhang**[1]     **Xiaolin Wu**[2*]
[1]Department of Electronic Engineering, Shanghai Jiao Tong University
[2]School of Computing and Artificial Intelligence, Southwest Jiaotong University
xzhang9308@gmail.com, xlw@swjtu.edu.cn

## Abstract

It is customary to deploy uniform scalar quantization in the end-to-end optimized Neural image compression methods, instead of more powerful vector quantization, due to the high complexity of the latter. Lattice vector quantization (LVQ), on the other hand, presents a compelling alternative, which can exploit inter-feature dependencies more effectively while keeping computational efficiency almost the same as scalar quantization. However, traditional LVQ structures are designed/optimized for uniform source distributions, hence nonadaptive and suboptimal for real source distributions of latent code space for Neural image compression tasks. In this paper, we propose a novel learning method to overcome this weakness by designing the rate-distortion optimal lattice vector quantization (OLVQ) codebooks with respect to the sample statistics of the latent features to be compressed. By being able to better fit the LVQ structures to any given latent sample distribution, the proposed OLVQ method improves the rate-distortion performances of the existing quantization schemes in neural image compression significantly, while retaining the amenability of uniform scalar quantization.

## 1 Introduction

Deep neural network (DNN) based image compression methods have quickly merged as the winner of rate-distortion performance over all their competitors, which is a remarkable achievement considering decades of slow progress in this heavily researched field. Their successes are mostly due to the capability of DNNs to learn compact and yet versatile latent representations of images. Nevertheless, all neural image compression systems will not be complete without the quantization and entropy coding modules for the learnt latent features.

Contrary to the sophistication of highly expressive non-linear latent image representations, quantizing the latent representations (the bottleneck of the autoencoder compression architecture) is done, in current prevailing practice, by uniform scalar quantizer. This design decision is apparently made in favor of the simplicity and computational efficiency. However, uniform scalar quantization is inevitably limited in capturing complex inter-feature dependencies present in the latent space, leading to lower coding efficiency, particularly for highly correlated or skewed source distributions.

In data compression, a common approach of overcoming the limitations of scalar quantization is vector quantization (VQ), in which a vector of tokens is quantized as a whole into a VQ codeword, allowing for effective exploitation of inter-feature dependencies. But designing optimal vector quantizers is an NP-hard problem. Moreover, the VQ encoding requires the expensive nearest neighbor search in high dimensional space. One way of balancing between the complexity and compression performance is to introduce some structures into the quantizers. Lattice vector quantization (LVQ) is such an approach.

---

*Corresponding author.

38th Conference on Neural Information Processing Systems (NeurIPS 2024).

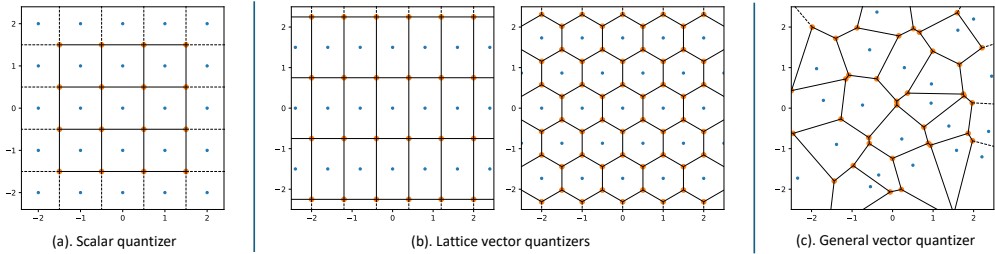

Figure 1: Two-dimensional visual examples for (a) scalar quantizer, (b) lattice vector quantizer and (c) general vector quantizer.

In LVQ, codewords are spatially organized into a regular high-dimensional lattice such that both LVQ encoding and decoding operations become almost as simple as in uniform quantization while achieving smaller quantization distortions.

Figure 1 contrasts the differences of three different types of quantizers: uniform scalar quantizer (USQ), lattice vector quantizer, and general free-form vector quantizer in two dimensions. As fully unconstrained VQ can have arbitrary Voronoi cell shapes, the VQ codewords can placed to best fit the source distributions among the three quantization schemes. But its lack of structures forces the VQ encoder to make a nearest neighbor decision that is expensive and not amenable to the end-to-end optimization of neural compression models. LVQ is a more efficient covering of the space than USQ and hence achieves higher coding efficiency; more importantly, the former enjoys the same advantages of the latter in easy and fast end-to-end DNN implementations [6].

In this paper, we propose a novel learning approach to adapt the LVQ geometry to the distributions of latent DNN features to be compressed, and improve the rate-distortion performance of existing LVQ designs for the tasks of compressing latent features of autoencoder architecture and neuron weights in general. The key contribution of our work is to abandon the traditional LVQ design goal of pursuing the partition of vector space with congruent quantizer cells whose shape is as close to hypersphere as possible. Instead, we propose a method to learn the bases of the LVQ generation matrix from sample statistics for minimum distortion. Such learnt bases are able to shape and orientate the quantizer cells to capture the correlations of latent features.

Through extensive experimentation on standard benchmark datasets, we demonstrate the effectiveness of our approach in significantly improving the compression performance of DNN-based image compression systems. Our method outperforms existing DNN quantization schemes in terms of both rate-distortion performance and computational complexity, increasing the cost-effectiveness of DNN image compression models.

In summary, this work advances the state-of-the-art of the end-to-end DNN image compression by learning the optimal shape and orientation of LVQ cells to exploit complex inter-feature dependencies for coding gains. The proposed learning approach of adapting LVQ codebooks to the underlying data distribution opens up a promising new path towards more resource efficient, practical and yet near-optimal solutions to the problem of DNN-based visual signal compression. Moreover, with minor adjustments, our work can be generalized to lossy compression of all other DNN models.

## 2 Related Work

Ballé *et al.* [6] pioneered the end-to-end CNN model for image compression, significantly advancing the field by integrating nonlinear transforms into the classical three-step signal compression framework: transform, quantization, and entropy coding. The key advantage of this CNN approach over traditional methods lies in replacing linear transforms with more powerful and complex nonlinear transforms, leveraging the distinctive capabilities of deep neural networks.

Following the groundbreaking work of Ballé *et al.*[6], numerous end-to-end compression methods have emerged, further enhancing rate-distortion (R-D) performance. These advancements have been achieved by introducing more sophisticated nonlinear transforms and more efficient context entropy models[38, 4, 22, 29]. Additionally, other works have focused on adaptive context models for entropy estimation, including [32, 7, 35, 26], with CNN methods by Minnen *etal.*[35, 26] outperforming

BPG in PSNR. Further, some studies have concentrated on content-adaptive approaches by updating encoder-side components during inference [41, 37, 28]. A substantial body of literature [13, 12, 30, 36, 33, 46, 20, 44, 17, 24, 47, 19, 21, 42] has also sought to improve R-D performance or coding efficiency. More recently, transformer-based image compression methods [50, 31] have been proposed as superior solutions over CNNs. Diffusion model-based image compression [45, 11] methods are also being explored by many researchers.

In most published end-to-end optimized DNN image compression methods, the quantization function typically adopts a uniform scalar quantizer rather than a vector quantizer. Few studies [3, 49, 48, 25, 16, 27] have delved into vector quantization in end-to-end image compression networks. Agustsson *etal.*[3] proposed a so-called soft-to-hard vector quantization module to bridge the nearest neighbor decision process of VQ and variational backpropagation necessary for end-to-end CNN training. Their technique is a soft (continuous) relaxation of the branching operation of VQ, allowing the VQ effects to be approximated in end-to-end training. Zhu *etal.*[49] proposed a probabilistic vector quantization with cascaded estimation to estimate means and covariances. However, the optimization goal in their work only considers the distortion term, overlooking the crucial design requirement of coding rates. More recently, Zhang *et al.* [48] suggested replacing scalar quantizers with a pre-defined lattice quantizer to construct an end-to-end image compression system, achieving improved rate-distortion performance with minimal increase in model complexity. To the best of our knowledge, this is the first work to investigate the learning of optimal lattice vector quantizers within end-to-end optimized DNN image compression systems.

## 3   Method

In this section, we introduce our method for integrating learnable lattice vector quantization (LVQ) into end-to-end optimized DNN image compression systems. We begin with a brief introduction of background on lattice vector quantization. Following this, the concept of a lattice and its mathematical properties are introduced. Next, we present our approach for differential lattice vector quantization, which enables the incorporation of LVQ into the backpropagation framework necessary for end-to-end training. We then describe the rate estimation based on the mixture of multivariate Gaussians, which is essential for entropy coding of the quantized features. Finally, we outline our method for learning the optimal lattice codebook through end-to-end training, ensuring that the quantization scheme is tailored to the specific characteristics of the image data.

### 3.1   Background

Lattice vector quantization (LVQ) [18, 1, 40] is a technique used in digital signal processing and data compression for efficient representation of data. It belongs to the broader family of vector quantization methods, which involves dividing a large set of vectors into smaller, representative clusters. In LVQ, the set of representative vectors is organized into a regular lattice structure within a multi-dimensional space. This lattice structure is defined by a set of basis vectors, known as lattice vectors, which form the nodes or points of the lattice. The LVQ process involves mapping input vectors from a high-dimensional space onto the nearest lattice points, effectively quantizing the input data by selecting the closest lattice point for each input vector. This quantized representation of the original data helps in reducing the amount of data while maintaining a high level of accuracy. In the context of image compression, LVQ offers significant advantages such as reduced computational complexity and memory requirements compared to other vector quantization methods.

### 3.2   Definition of Lattice

A *lattice* $\Lambda$ in $\mathbb{R}^n$ is a discrete subgroup of $\mathbb{R}^n$ that spans the space $\mathbb{R}^n$ as a vector space over $\mathbb{R}$. Mathematically, a lattice can be expressed as:

$$\mathbf{\Lambda} = \{\boldsymbol{x} \in \mathbb{R}^n \mid \boldsymbol{x} = \sum_{i=1}^{n} \boldsymbol{b}_i m_i, \, m_i \in \mathbb{Z}\}, \tag{1}$$

where $\boldsymbol{b}_1, \boldsymbol{b}_2, \ldots, \boldsymbol{b}_n$ are linearly independent vectors in $\mathbb{R}^n$, called the basis vectors of the lattice. In other words, a vector $\boldsymbol{x}$ is a lattice point if it can be formed as a linear combination of the basis vectors scaled by integers $m_i \in \mathbb{Z}$.

The matrix-form representation of the above equation is:

$$\Lambda = \{ \boldsymbol{x} \in \mathbb{R}^n \mid \boldsymbol{x} = \boldsymbol{B}\boldsymbol{m}, \boldsymbol{m} \in \mathbb{Z}^n \}, \tag{2}$$

where $\boldsymbol{B}$ is an $n \times n$ matrix called the basis (or generator) matrix:

$$\boldsymbol{B} = \begin{bmatrix} | & | & & | \\ \boldsymbol{b}_1 & \boldsymbol{b}_2 & \cdots & \boldsymbol{b}_n \\ | & | & & | \end{bmatrix}. \tag{3}$$

It can be observed that a lattice $\Lambda$ is fully determined by its basis matrix $\boldsymbol{B}$. Therefore, learning a lattice codebook is equivalent to learning the basis matrix $\boldsymbol{B}$.

The Voronoi cell of a lattice point $\boldsymbol{x} \in \Lambda$ is defined as the set of points closer to $\boldsymbol{x}$ than to any other lattice points:

$$V(\boldsymbol{x}) \triangleq \{ \boldsymbol{z} \in \mathbb{R}^n \mid \|\boldsymbol{z} - \boldsymbol{x}\| \le \|\boldsymbol{z} - \hat{\boldsymbol{x}}\|, \forall \hat{\boldsymbol{x}} \in \Lambda \}. \tag{4}$$

The LVQ codewords are the centroids of the Voronoi polyhedrons, which are of the same size.

### 3.3 Differential lattice vector quantization

Given a lattice $\Lambda$ and any vector $\boldsymbol{v} \in \mathbb{R}^n$ to be quantized, applying lattice vector quantization $\boldsymbol{q}_l$ on $\boldsymbol{v}$ involves finding the lattice point $\boldsymbol{q}_l(\boldsymbol{v})$ in $\Lambda$ that is closest to $\boldsymbol{v}$. This can be mathematically expressed as:

$$\|\boldsymbol{q}_l(\boldsymbol{v}) - \boldsymbol{v}\| \le \|\boldsymbol{x} - \boldsymbol{v}\|, \quad \forall \boldsymbol{x} \in \Lambda. \tag{5}$$

This is known as the classical closest vector problem (CVP) [34, 2], which is NP-complete. Additionally, it is not differentiable, and thus cannot be directly integrated into the end-to-end learning scheme.

To address this challenge, we propose using Babai's Rounding Technique (BRT) [5] to estimate a vector that is sufficiently close to $\boldsymbol{v}$, although it may not be the exact closest lattice point to $\boldsymbol{v}$. Specifically, according to BRT, any given vector $\boldsymbol{v}$ can be quantized to a sufficiently close lattice point by:

$$\boldsymbol{q}_l(\boldsymbol{v}) = \boldsymbol{B}\lfloor \boldsymbol{B}^{-1}\boldsymbol{v} \rceil, \tag{6}$$

where $\lfloor \cdot \rceil$ represents the rounding operation. During training, the rounding operation is replaced by adding uniform noise to enable integration into end-to-end optimization.

However, our experiments revealed that the BRT-estimated lattice vector may be far away from the exact closest lattice point if the cell shape of the optimized LVQ is arbitrary and unconstrained. This discrepancy causes an inconsistency between training and inference, if the learnt lattice vector quantizer is employed in the inference stage. As analyzed in [5], the BRT-estimated lattice point will approach the closest lattice point if the basis vectors of the generator matrix are mutually orthogonal. For this reason, we propose to impose orthogonal constraints on the basis vectors of the generator matrix, enhancing the accuracy of the BRT-estimated lattice point and reducing the gap between lattice vector quantization during training and inference.

The orthogonality constraint on the basis vectors $\{ \boldsymbol{b}_i, i = 1, \cdots, n \}$ can be expressed as:

$$\boldsymbol{b}_i^\top \cdot \boldsymbol{b}_j = 0, \quad \forall i \ne j, \tag{7}$$

where $\boldsymbol{b}_i$ and $\boldsymbol{b}_j$ are the basis vectors in the basis matrix $\boldsymbol{B}$. By imposing orthogonal constraints on the basis vectors, we can enhance the accuracy of the estimates provided by Babai's Rounding Technique (BRT) and improve training stability. This approach effectively reduces errors in practical applications, providing more reliable quantization results.

According to [23], the initialization of lattice basis matrix is also important for stable training. To this end, we propose to give a good initialization of the basis matrix, by uniformly initializing $\boldsymbol{B}$ as in the following equation:

$$\boldsymbol{B} \sim U(-\frac{1}{\sqrt[n]{S} - 1}, \frac{1}{\sqrt[n]{S} - 1}) \tag{8}$$

where $S$ is the desired codebook size of learned lattice vector quantizers. The above range is derived by assuming idealized quantization on a set of linearly independent dimensions, providing us with a initializing point for the lattice density.

## 3.4  Rate estimation

In the proposed LVQ-based end-to-end image compression system, the latent features are quantized to the nearest lattice vector point and then compressed using entropy coding techniques, such as arithmetic coding [39]. Entropy coding of the lattice vectors requires estimating their (conditional) probability distributions.

In end-to-end image compression methods, a mixture of Gaussians is commonly used to model the (conditional) probability distribution of latent features. Extending this approach, we propose to model the probability distribution for the lattice vectors using a mixture of $n$-dimensional multivariate Gaussians [49]:

$$p_{\boldsymbol{m}} = \int_{V(m)} \sum_{k=1}^{K} \boldsymbol{\Phi_k} \mathcal{N}(x; \mu_k, \Sigma_k) dx, \tag{9}$$

$$\Phi \sim \text{Categorical}(K, \phi), \tag{10}$$

where $\mu_k$ and $\Sigma_k$ are the mean and covariance matrix of the $k$-th Gaussian component, respectively. $\Phi$ represents the mixture coefficients, $V(m)$ is the area of Voronoi polyhedron.

Computing the integral in the equations above presents a significant numerical challenge, primarily because the integration region, $V(m)$, forms a complex polytope in most lattice structures. Additionally, as dimensionality increases, the computational complexity of evaluating these integrals becomes impractical. To address this challenge, Kudo *etal.* [25] proposed an approach using Monte Carlo (MC) estimation to approximate these irregular integrals. Specifically, they apply a Quasi-Monte Carlo (QMC) method, which leverages low-discrepancy sequences, such as Faure sequences, for sampling. The QMC method offers a key advantage in convergence rate over the conventional MC method, achieving nearly $O(1/M)$ convergence as opposed to the slower $O(1/\sqrt{M})$ convergence rate of traditional MC methods. However, since the MC/QMC method contains estimation error, which may lead to the encoding/decoding failure during the inference stage.

We adopt another approach to overcome the difficulty of integration over a irregular region. As discussed in Section 3.3, to reduce the gap between lattice vector quantization during training and inference, we apply orthogonal constraints on the basis vectors of the generator matrix $B$. Given that the basis vectors are approximately orthogonal, the joint probability distribution of a lattice vector point can be approximately decomposed into the product of the probability distributions of each independent variable in the vector. Assuming that each variable in the vector $\boldsymbol{m}$ is modeled by a mixture of univariate Gaussians, Equation 10 can be rewritten as:

$$p_{\boldsymbol{m}} = \int_{V(m)} \sum_{k=1}^{K} \boldsymbol{\Phi_k} \mathcal{N}(x; \mu_k, \Sigma_k) dx \approx \prod_{i=1}^{n} \int_{V(m^i)} \sum_{k=1}^{K} \phi_{\mathbf{k}}{}^{i} \mathcal{N}(x^i; \mu_k^i, \sigma_k^i) dx^i. \tag{11}$$

The parameters $\phi_{\mathbf{k}}{}^{i}$, $\mu_k^i$ and $\sigma_k^i$ are predicted by a neural network which is trained along with the entire compression system. Given the predicted probability distribution in Equation 11, the estimated rate of the lattice vector can be expressed as:

$$R = \mathbb{E}_{x \sim X}[-log_2 p_{\boldsymbol{m}}(x)] = \mathbb{E}_{x \sim X}[-log_2 \prod_{i=1}^{n} \int_{V(m^i)} \sum_{k=1}^{K} \phi_{\mathbf{k}}{}^{i} \mathcal{N}(x^i; \mu_k^i, \sigma_k^i) dx^i]. \tag{12}$$

By decomposing the joint distribution in this manner, we significantly reduce the computational complexity, making the method more practical and efficient. This decomposition leverages the orthogonality of the basis vectors, ensuring that the joint probability can be approximated by the product of marginal probabilities, thus enhancing the feasibility of entropy estimation in high-dimensional spaces.

In addition, for lattice vector quantization, it essentially involves finding the integer combination coefficients of the basis vectors that correspond to the closest lattice point. Therefore, coding the quantized features (lattice vector) is equivalent to coding the integer combination coefficients (indexes) of the basis vectors. We used the latter in our experiments due to its convenience.

## 3.5 Learning lattice codebook via end-to-end training

In end-to-end optimized DNN image compression methods, an image vector $I \in \mathbb{R}^N$ is mapped to a latent code space via a parametric nonlinear analysis transform, $Y = g_a(I; \theta_g)$. This representation is then quantized by lattice vector quantizer, yielding a discrete-valued vector $\hat{Y} = q_l(Y) \in \mathbb{Z}^M$, which can be losslessly compressed using entropy coding algorithms such as arithmetic coding [39, 43] and transmitted as a sequence of bits. At the decoder side, $\hat{Y}$ is recovered from the compressed codes and subjected to a parametric nonlinear synthesis transform $g_s(\hat{Y}; \phi_g)$ to reconstruct the image $\hat{I}$. The nonlinear parametric transforms $g_a(I; \theta_g)$ and $g_s(\hat{Y}; \phi_g)$ are implemented using convolutional neural networks, with parameters $\theta_g$ and $\phi_g$ learned from a large dataset of natural images.

The training objective is to minimize both the expected length of the bitstream and the expected distortion of the reconstructed image relative to the original, resulting in a rate-distortion-orthogonality optimization problem:

$$
\begin{aligned}
& \text{minimize} \quad R + \lambda_1 \cdot D + \lambda_2 \cdot L \\
& R = \mathbb{E}_{I \sim p_I}[-\log_2 p_{\hat{Y}}(q_l(g_a(I)))] \\
& D = \mathbb{E}_{I \sim p_I}[d(I, g_s(q_l(g_a(I))))] \\
& L = \sum_{i,j=1}^{n} |b_i^\top \cdot b_j|, \forall i \neq j
\end{aligned}
\tag{13}
$$

where $p_I$ is the (unknown) distribution of source images, and $p_{\hat{Y}}$ is a discrete entropy model. $\lambda_1$ is the Lagrange multiplier that determines the desired rate-distortion trade-off and $\lambda_2$ is Lagrange multiplier for the regularization term of orthogonal constraints on the basis vectors. $q_l(\cdot)$ represents lattice vector quantization as described in Equation 6. The first term (rate) corresponds to the cross-entropy between the marginal distribution of the latent variables and the learned entropy model, which is minimized when the two distributions match. The second term (distortion) measures the reconstruction error between the original and reconstructed images. The third term controls the orthogonality of the basis vectors for the learned lattice codebook.

## 4 Experiments

To evaluate the effectiveness of our proposed lattice vector quantization method for end-to-end image compression, we conducted a series of experiments on widely-used image datasets. This section details the experimental setup, including the network architectures, datasets, training details, evaluation metrics, etc. We also provide a comprehensive analysis of the performance of our method, highlighting its advantages in terms of rate-distortion performance and computational efficiency.

### 4.1 Experiment setup

In this part, we describe the experiment setup including the following four aspects: network architecture, context model, training and evaluation.

**Network**. We aim to evaluate the superiority of the proposed OLVQ method over the scalar quantizer across models with varying complexities. To this end, we selected three well-known networks that are considered milestones in end-to-end image compression: Bmshj2018 [7], SwinT-ChARM [50], and LIC-TCM [31]. These models range in complexity from low to high, allowing us to assess the adaptability of the LVQ to different levels of model complexity and provide a more comprehensive comparison.

**Context model**. In addition to model complexity, we also aim to evaluate the adaptability of LVQ to different context models. Specifically, we test four common context models varying in complexity from low to high: Factorized [7], Checkerboard [20], Channel-wise Autoregressive [36], and Spatial-wise Autoregressive [35]. By systematically testing the LVQ against these context models, we can determine its effectiveness in enhancing compression efficiency and adaptability. This evaluation will help us understand the benefits and limitations of LVQ in both simple and complex settings, providing valuable insights into its practical applicability in real-world image compression tasks.

**Training**. The training dataset comprises high-quality images carefully selected from the ImageNet dataset [15]. To ensure a robust and challenging training process, we filter the images from ImageNet

to include only those with a resolution exceeding one million pixels. This ensures that the dataset consists of detailed and complex images, which are crucial for training effective image compression models. Training images are then random-cropped to $256 \times 256$ and batched into 16. This crop-based approach not only maximizes the utilization of each high-resolution image but also introduces variability, which is crucial for robust training.

We undertake the training of all combinations of the three selected networks and the four context models from scratch. Each network and context model combination is trained for 3 million iterations. We train each model using the Adam optimizer with $\beta-1 = 0.9, \beta_2 = 0.999$. The initial learning rate is set to $10^{-4}$ for the first 2M iterations, and then decayed to $10^{-5}$ for another 1M iterations training. All modules including the proposed learnable lattice vector quantization modules are implemented in PyTorch and CompressAI [9]. All experiments are conducted with four RTX 3090 GPUs.

In general, the latent feature of an image is represented as a cube with dimensions $H \times W \times C$. We choose to group features by channels instead of spatial dimensions, as features across different channels at the same spatial location exhibit strong correlations, making them well-suited for lattice vector quantization. The dimension of each vector is set to 32. Consequently, the quantized output of a latent feature with an initial size of $H \times W \times C$ will have a size of $H \times W \times \frac{C}{32}$, resulting in a thicker cube with the same spatial dimensions but fewer channels.

**Evaluation**. The trained compression models are evaluated on two widely used datasets: the Kodak dataset [14] and the CLIC validation set [8]. These datasets are chosen for their diversity and representativeness of real-world image content, making them ideal benchmarks for assessing the performance of image compression algorithms. We evaluate the different models using the Bjøntegaard delta rate (BD-rate) [10], a metric that quantifies the average bitrate savings for the same level of reconstruction quality. The BD-rate provides a comprehensive measure of compression efficiency by comparing the rate-distortion curves of different methods. For each model, the BD-rate is computed for every image in the datasets, ensuring a thorough assessment of performance. The individual BD-rate values are then averaged across all images within each dataset, offering a robust indicator of overall compression efficiency. By utilizing BD-rate, we can objectively compare the bitrate savings achieved by our proposed optimal lattice vector quantizer (LVQ) method against traditional scalar quantizers across models of varying complexities coupled with different context models. This evaluation helps to highlight the strengths and potential limitations of our approach in practical scenarios.

## 4.2 Comparison with scalar quantizer

Table 1 presents the bitrate savings achieved by the proposed optimal lattice vector quantization (LVQ) method over the scalar quantizer across different image compression networks and context models. Based on the results, we can make several observations:

**Overall Performance Trends**. Across all context models and network configurations, the OLVQ method consistently achieves bitrate savings over the scalar quantizer. Besides, the effectiveness of LVQ decreases as the complexity of the context model increases.

**Performance Across Networks**. For Bmshj2018, the OLVQ method achieves the highest bitrate savings with this network, ranging from -22.60% with the Factorized context model to -8.31% with the Spatial-wise Autoregressive model. For SwinT-ChARM, the savings are more modest compared to Bmshj2018, with the highest savings of -12.44% for the Factorized context model and the lowest of -2.31% for the Spatial-wise Autoregressive model. For LIC-TCM, this network shows the least savings, with the highest at -8.51% (Factorized) and the lowest at -0.95% (Spatial-wise Autoregressive).

**Impact of Context Models**. The Factorized context model consistently shows the highest bitrate savings across all networks. This suggests that simpler context models benefit more from the OLVQ method. The Checkerboard and Channel-wise Autoregressive models show intermediate savings, indicating that while LVQ is beneficial, the gains are reduced as the context model becomes more complex. The Spatial-wise Autoregressive model, being the most complex, shows the least bitrate savings. This indicates that the benefits of LVQ diminish as the context model complexity increases.

In summary, the proposed OLVQ method significantly improves bitrate savings over scalar quantization, especially in simpler networks and context models. However, its effectiveness diminishes with

Table 1: Results of bitrate savings achieved by the proposed optimal lattice vector quantization (OLVQ) method over the scalar quantizer across different image compression networks and context models. The lattice vector dimension is set to 32 here.

| Network
Context model | Bmshj2018 [7] | SwinT-ChARM [50] | LIC-TCM [31] |
|---|---|---|---|
| Factorized (w/o context) [7] | -22.60% | -12.44% | -8.51% |
| Checkerboard [20] | -11.94% | -6.37% | -2.18% |
| Channel-wise Autoregressive [36] | -10.61% | -5.61% | -2.03% |
| Spatial-wise Autoregressive [35] | -8.31% | -2.31% | -0.95% |

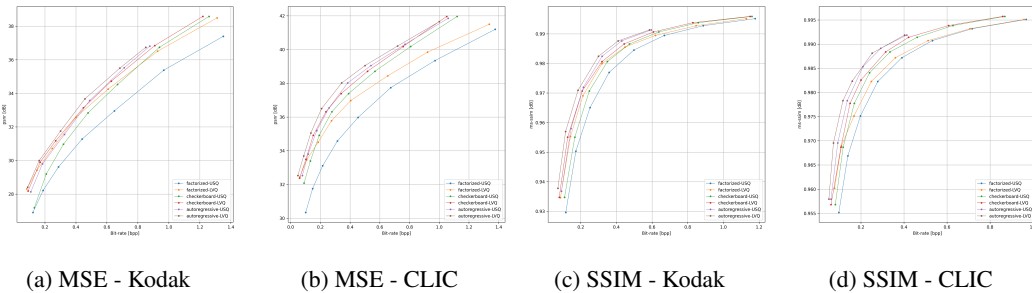

(a) MSE - Kodak    (b) MSE - CLIC    (c) SSIM - Kodak    (d) SSIM - CLIC

Figure 2: Rate-distortion curves for the Bmshj2018 model on the Kodak and CLIC datasets, evaluated in terms of MSE and SSIM.

the increasing complexity of the context models and networks. This indicates that while LVQ is a powerful tool for improving compression efficiency, its benefits are more pronounced in less complex settings. This insight can guide the choice of quantization methods based on the complexity of the image compression model and context model being used.

We also provide R-D curves for the Bmshj2018 model on the Kodak and CLIC datasets in Fig. 2. It can be observed from these R-D curves that the proposed optimal lattice vector quantization consistently improves the existing compression models in both low and high bitrate cases. However, the performance gain is more pronounced at lower bitrates. It can also be observed that the proposed OLVQ method coupled with the factorized entropy model can approach the performance of the autoregressive entropy model, particularly at low bitrates. For high bitrates, the autoregressive entropy model still has an edge. Given this, we believe that the proposed OLVQ method can serve as a viable alternative to the autoregressive entropy model, especially in scenarios where low bitrate performance is crucial and computational efficiency is a priority. The factorized entropy model, being less computationally intensive, combined with our OLVQ method, offers an attractive trade-off between performance and efficiency.

### 4.3 Comparison with non-learned lattice vector quantizers

In addition to the comparison with scalar quantizer, we also conduct comarisons agasin the classical non-learned lattice vector quantizers, such as Gosset lattice (E8), Barnes-Wall lattice (A16) and Leech lattice (A24). Table 2 tabulates the bitrate savings achieved by classical pre-defined lattice vector quantizers and the learned optimal lattice vector quantizers over the scalar quantizer, across different image compression models (Bmshj2018 and SwinT-ChARM) and context models (Factorized and Checkerboard). Based on the results, several observations can be made:

**Overall Performance Comparison**. The learned optimal lattice vector quantizers consistently outperform the classical pre-defined lattice vector quantizers across all configurations, which demonstrates the effectiveness of learning the lattice structures tailored to specific (unknown) source distributions of latent space.

**Impact of Lattice Dimensions**. The optimized 8-dimension lattice shows significant improvements over the Gosset lattice (E8). The highest improvement is seen in the Bmshj2018 model with the Factorized context, achieving -10.92% compared to -6.58%. For 16-dimension, the optimized lattice

Table 2: Results of bitrate savings achieved by classical pre-defined lattice vector quantizers and the learned optimal lattice vector quantizers over the scalar quantizer, across different image compression models and context models.

| Auto-encoder size | Bmshj2018 [7] | | SwinT-ChARM [50] | |
|---|---|---|---|---|
| Context model | Factorized | Checkerboard | Factorized | Checkerboard |
| E8, Gosset lattice | -6.58% | -4.95% | -5.12% | -2.19% |
| Optimized 8-dimension lattice | 10.92% | -6.47% | -6.03% | -3.02% |
| A16, Barnes-Wall lattice | -8.15% | -6.15% | -5.23% | -3.12% |
| Optimized 16-dimension lattice | -15.73% | -8.82% | -7.42% | -4.64% |
| A24, Leech lattice | -12.82% | -7.54% | -6.69% | -3.47% |
| Optimized 24-dimension lattice | -20.49% | -10.71% | -10.68% | -5.89% |

provides substantial bitrate savings over the Barnes-Wall lattice (A16). Notably, for Bmshj2018 with Factorized context, the optimized lattice achieves -15.73%, a marked improvement from -8.15%. For 24-dimension, the optimized lattice shows the greatest bitrate savings, particularly in the Bmshj2018 model. The highest improvement is observed in the Bmshj2018 model with Factorized context, with savings of -20.49% compared to -12.82%.

**Impact of Context Models**. Factorized context models generally exhibit greater savings than Checkerboard contexts. This trend is evident in both pre-defined and optimized lattices, indicating that simpler context models may benefit more from lattice optimization. The largest relative improvement by the optimized lattice over the classical lattice is seen with Factorized context, suggesting that the gains from learning the lattice structure are more pronounced in simpler context settings.

In summary, the learned optimal lattice vector quantizers demonstrate substantial improvements in bitrate savings over classical pre-defined lattice quantizers. The benefits are more pronounced in higher-dimensional lattices and simpler context models. This trend suggests that higher-dimensional optimized lattices are more adaptable and efficient, providing superior performance across different models and context configurations.

## 4.4 Comparison with general vector quantizers

We conduct comparisons between the lattice vector quantizers (LVQ) and the general vector quantizers (GVQ) and report the bitrate savings over scalar quantizer in Table 3.

Table 3: Results of bitrate savings achieved by the learned optimal lattice vector quantizers and general vector quantizers over uniform scalar quantizer, across different image compression models and context models.

| Auto-encoder size | Bmshj2018 [7] | | SwinT-ChARM [50] | |
|---|---|---|---|---|
| Context model | Factorized | Checkerboard | Factorized | Checkerboard |
| Optimized 8-dimension LVQ | -10.92% | -6.47% | -6.03% | -3.02% |
| Optimized 8-dimension GVQ | -12.10% | -7.82% | -7.59% | -4.19% |
| Optimized 16-dimension LVQ | -15.73% | -8.82% | -7.42% | -4.64% |
| Optimized 16-dimension GVQ | -17.59% | -10.48% | -9.41% | -6.03% |
| Optimized 24-dimension LVQ | -20.49% | -10.71% | -10.68% | -5.89% |
| Optimized 24-dimension GVQ | -22.74% | -12.80% | -13.15% | -8.28% |

It can be seen that, for both Bmshj2018 and SwinT-ChARM models, the optimized lattice vector quantizer is approaching the perfromance of the optimized general vector quantizer (GVQ) in terms of bitrate savings. This trend is observed across all dimensions (8, 16, and 24) and context models (Factorized and Checkerboard). Besides, increasing the dimension of the quantizer (from 8 to 24) leads to greater bitrate savings for both LVQ and GVQ. This is consistent across all models and context types. For instance, the bitrate savings for the Bmshj2018 model with a Factorized context model improves from -10.92% (8-dimension LVQ) to -20.49% (24-dimension LVQ), and from -12.10% (8-dimension GVQ) to -22.74% (24-dimension GVQ).

## 4.5 Inference time

In this subsection, we compare the inference times of three different quantization methods: uniform scalar quantizer (USQ), lattice vector quantizer (LVQ), and general vector quantizer (GVQ). Table 4 tabulates the inference times of different quantizers across varying dimensions.

Table 4: Inference times of three different quantization methods: uniform scalar quantizer (USQ), lattice vector quantizer (LVQ), and general vector quantizer (GVQ) across different dimensions. For GVQ, the codebook size is adjusted to be larger as the dimension increases.

| Quantizer \ Dimension | 1 | 8 | 16 | 32 |
|---|---|---|---|---|
| Uniform scalar quantizer (USQ) | $\sim$ 5ms | / | / | / |
| Lattice vector quantizer (LVQ) | / | $\sim$ 18ms | $\sim$ 22ms | $\sim$ 40ms |
| General vector quantizer (GVQ) | / | $\sim$ 123ms | $\sim$ 216ms | $\sim$ 454ms |

First, as expected, the USQ demonstrates extremely low inference time (approximately 5ms), reflecting its simplicity and efficiency. The LVQ shows moderate inference times across the evaluated dimensions. For an 8-dimensional quantization, LVQ takes about 18ms, increasing to 40ms for a 32-dimensional quantization. This increase in time is relatively modest compared to GVQ, indicating that LVQ scales more efficiently with dimension size. In contrast, the GVQ incurs significantly higher inference times, particularly as the dimension increases. At 8 dimensions, GVQ takes around 123ms, which escalates to 454ms at 32 dimensions. This substantial increase highlights the computational complexity associated with GVQ, especially with larger codebook sizes required for higher dimensions. Overall, the LVQ provides a good balance between inference time and scalability. While not as fast as USQ, LVQ's inference times remain manageable and much lower than those of GVQ, particularly for higher-dimensional data. This makes LVQ a viable option for applications requiring efficient and scalable vector quantization.

## 4.6 Limitation

While the proposed OLVQ method demonstrates significant improvements in rate-distortion performance for DNN-based image compression, one limitation must be acknowledged: The proposed method is particularly well-suited for lightweight end-to-end image compression models that adopt inexpensive context models. These models typically benefit more from the enhanced exploitation of inter-feature dependencies provided by LVQ. For the heavy-duty compression models that employ complex auto-encoders and sophisticated context models, the performance gains achieved by the proposed OLVQ method are relatively modest.

## 5 Conclusion

In this paper, we addressed the limitations of traditional lattice vector quantization (LVQ) in the context of end-to-end DNN image compression. While LVQ offers the potential to more effectively exploit inter-feature dependencies compared to uniform scalar quantization, its design has historically been suboptimal for real-world distributions of latent code spaces. To overcome this, we introduced a novel learning method that designs rate-distortion optimal LVQ codebooks tailored to the sample statistics of the latent features. Our proposed method successfully adapts the LVQ structures to any given latent sample distribution, significantly improving the rate-distortion performance of existing quantization schemes in DNN-based image compression. This enhancement is achieved without compromising the computational efficiency characteristic of uniform scalar quantization. Our results demonstrate that by better fitting LVQ structures to actual data distributions, one can achieve superior compression performance. paving the way for more efficient and effective image compression techniques in practical applications.

## Broader Impact

The proposed OLVQ method not only advances the state of image compression technology but also offers practical benefits across multiple industries and for end-users. Its ability to improve efficiency, reduce costs, and enhance user experiences underscores its potential for making a positive and substantial impact on both technology and society.

## Acknowledgment

This work was supported in part by the National Natural Science Foundation of China (No.62301313) and China Postdoctoral Science Foundation funded project (No.2024T170559).

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

# A Appendix / supplemental material

## A.1 Comparison with other VQ based image compression methods

We provide the comparison results with the following vector quantization-based image compression methods: SHVQ (NeurIPS '17) and McQUIC (CVPR '22). From the provided table, it can be observed that the proposed OLVQ method not only outperforms these VQ-based image compression methods in terms of BD-rate but also demonstrates superior performance in computational complexity. As shown, our method achieves lower BD-rates, i.e., better compression performance. At the same time, the computational complexity of our approach is significantly lower than SHVQ and McQUIC, making it more practical, especially on resource-constrained devices.

| Quantizers | Bmshj2018 Factorized | Bmshj2018 Checkerboard | SwinT-ChARM Factorized | SwinT-ChARM Checkerboard |
|---|---|---|---|---|
| OLVQ | -20.49% | -10.71% | -10.68% | -5.89% |
| SHVQ [1] | -8.23% | -5.18% | -6.55% | -1.92% |
| McQUIC [2] | -15.11% | -7.93% | -7.24% | -3.18% |

## A.2 Ablation study of orthogonal constraint

We propose to impose orthogonal constraints on the basis vectors of the generator matrix, enhancing the accuracy of the BRT-estimated lattice point and reducing the gap between lattice vector quantization during training and inference.

**The performance without the orthogonal constraint will drop by about 4.2% for BD-rate.** We report the detailed performance numbers with and without the orthogonal constraint in the following table. The rate-distortion curves are also provided in the rebuttal PDF.

| Entropy Model | Bmshj2018 w/ orthogonal constraint | Bmshj2018 w/o orthogonal constraint | SwinT-ChARM w/ orthogonal constraint | SwinT-ChARM w/o orthogonal constraint |
|---|---|---|---|---|
| Factorized (w/o context) | -22.60% | -18.2% | -12.44% | -8.09% |
| Checkerboard | -11.94% | -7.94% | -6.37% | -2.25% |
| Channel-wise Autoregressive | -10.61% | -6.34% | -5.61% | -1.22% |
| Spatial-wise Autoregressive | -8.31% | -4.20% | -2.31% | +2.08% |

## A.3 The impact of $\lambda_2$ and the term L in the loss function

We conduc ablation studies to thoroughly examine how these two hyperparameters ($\lambda_2$ and L) affect the performance of our method.

- For $\lambda_2$, we employed a search strategy to identify the optimal value that yields the best performance. Our findings indicate that both excessively large and excessively small values of $\lambda_2$ can lead to performance degradation. When $\lambda_2$ is too large, it overly constrains the shape of the optimized lattice codebook, resulting in a sub-optimal lattice structure. Conversely, when $\lambda_2$ is too small, it loosens the orthogonal constraint, causing a significant gap between lattice vector quantization during training and inference (Please refer to the global rebuttal for more details).

- Regarding the loss term L, our initial experiments utilized the MSE metric. To further investigate, we have now included a study using the SSIM metric as the loss term L. The results from this study indicate that our conclusions drawn from using the MSE metric remain valid when using the SSIM metric. Specifically, the performance trends and the effectiveness of our proposed method are consistent across both metrics.

### A.4 Babai's Rounding Algorithm and Its Bound

#### A.4.1 Mathematical Formulation

Given a lattice $\Lambda$ with a basis $B = [\mathbf{b}_1, \mathbf{b}_2, \ldots, \mathbf{b}_n]$, we want to find a lattice point $\mathbf{v} \in \Lambda$ that is close to a target vector $\mathbf{t} \in \mathbb{R}^n$.

Babai's rounding algorithm involves the following steps:

1. Compute the Gram-Schmidt orthogonalization of the basis $B$, resulting in an orthogonal basis $B^* = [\mathbf{b}_1^*, \mathbf{b}_2^*, \ldots, \mathbf{b}_n^*]$.

2. Express the target vector $\mathbf{t}$ in terms of the orthogonal basis $B^*$:

$$\mathbf{t} = \sum_{i=1}^{n} c_i \mathbf{b}_i^*$$

   where $c_i$ are the coordinates of $\mathbf{t}$ in the Gram-Schmidt basis $B^*$.

3. Round each coordinate $c_i$ to the nearest integer:

$$\mathbf{c}' = (\mathrm{round}(c_1), \mathrm{round}(c_2), \ldots, \mathrm{round}(c_n))$$

4. Construct the approximate lattice point:

$$\mathbf{v} = \sum_{i=1}^{n} \mathrm{round}(c_i) \mathbf{b}_i$$

#### A.4.2 Example

Suppose we have a 2-dimensional lattice with basis vectors $b_1$ and $b_2$, and a target vector $\mathbf{t}$.

1. **Compute the Gram-Schmidt basis:**

$$b_1^* = b_1$$

$$b_2^* = b_2 - \frac{\langle b_2, b_1 \rangle}{\|b_1\|^2} b_1$$

2. **Project $\mathbf{t}$ onto $b_1^*$ and $b_2^*$:**

$$c_1 = \frac{\langle \mathbf{t}, b_1^* \rangle}{\|b_1^*\|^2}$$

$$c_2 = \frac{\langle \mathbf{t}, b_2^* \rangle}{\|b_2^*\|^2}$$

3. **Round $c_1$ and $c_2$:**

$$\mathrm{round}(c_1), \mathrm{round}(c_2)$$

4. **Construct the approximate lattice point:**

$$\mathbf{v} = \mathrm{round}(c_1) b_1 + \mathrm{round}(c_2) b_2$$

#### A.4.3 Proof of the Bound

We want to show that the distance between the target vector $\mathbf{t}$ and the found lattice point $\mathbf{v}$ is bounded by half the sum of the lengths of the basis vectors.

1. **Decompose the Target Vector:**

$$\mathbf{t} = \sum_{i=1}^{n} c_i \mathbf{b}_i^*$$

   Let $\mathbf{t}'$ be the projection of $\mathbf{t}$ onto the lattice:

$$\mathbf{t}' = \sum_{i=1}^{n} \mathrm{round}(c_i) \mathbf{b}_i^*$$

The difference between $\mathbf{t}$ and $\mathbf{t}'$ is:

$$\mathbf{t} - \mathbf{t}' = \sum_{i=1}^{n}(c_i - \text{round}(c_i))\mathbf{b}_i^*$$

Each term $c_i - \text{round}(c_i)$ is at most $\frac{1}{2}$ in magnitude because $\text{round}(c_i)$ is the nearest integer to $c_i$.

2. **Bound the Difference**:

$$\|\mathbf{t} - \mathbf{t}'\| \leq \sum_{i=1}^{n}|c_i - \text{round}(c_i)|\|\mathbf{b}_i^*\|$$

Since $|c_i - \text{round}(c_i)| \leq \frac{1}{2}$, we have:

$$\|\mathbf{t} - \mathbf{t}'\| \leq \sum_{i=1}^{n}\frac{1}{2}\|\mathbf{b}_i^*\| = \frac{1}{2}\sum_{i=1}^{n}\|\mathbf{b}_i^*\|$$

3. **Reconstruct the Lattice Point**: The lattice point $\mathbf{v}$ constructed by Babai's rounding algorithm is:

$$\mathbf{v} = \sum_{i=1}^{n}\text{round}(c_i)\mathbf{b}_i$$

Note that $\mathbf{t}'$ is the projection in the Gram-Schmidt basis, while $\mathbf{v}$ is the actual lattice point in the original basis. Since $\mathbf{v}$ and $\mathbf{t}'$ differ only by the orthogonalization process, the distance bound remains valid.

4. **Final Bound**: Since the Gram-Schmidt process does not increase the lengths of the basis vectors, we can state that:

$$\|\mathbf{t} - \mathbf{v}\| \leq \frac{1}{2}\sum_{i=1}^{n}\|\mathbf{b}_i\|$$

Thus, Babai's rounding algorithm guarantees that the distance between the target vector $\mathbf{t}$ and the found lattice point $\mathbf{v}$ is within half the sum of the lengths of the basis vectors.

## A.5 Broader Impact

The proposed Optimal Lattice Vector Quantization (LVQ) method for image compression offers several significant and wide-reaching benefits.

Firstly, LVQ leads to more efficient storage and transmission of images, which is crucial for industries that rely heavily on image data, such as digital media, medical imaging, and remote sensing. By reducing the file sizes without compromising image quality, LVQ helps in lowering storage costs and bandwidth requirements. This efficiency not only results in cost savings but also reduces the environmental impact by decreasing the energy consumption associated with data storage and transmission infrastructure.

Secondly, the enhanced image compression achieved through LVQ can result in faster image loading times and reduced latency in applications like web browsing, video streaming, and online gaming. Improved compression translates to quicker data transfer and reduced buffering, significantly enhancing the user experience. This is particularly beneficial in regions with limited bandwidth or high-latency networks, where efficient data compression can make a substantial difference in accessibility and performance.

Furthermore, the advancements in image compression technology provided by LVQ have broader implications for various fields. In digital media, for example, it enables higher quality streaming services and better user experiences. In medical imaging, it facilitates the efficient storage and sharing of high-resolution images, which are critical for accurate diagnosis and treatment. In remote sensing, improved compression allows for faster processing and analysis of satellite images, aiding in timely decision-making and response in fields such as disaster management and environmental monitoring.

Overall, the proposed LVQ method not only advances the state of image compression technology but also offers practical benefits across multiple industries and for end-users. Its ability to improve efficiency, reduce costs, and enhance user experiences underscores its potential for making a positive and substantial impact on both technology and society.

