# OpenReview forum: "Learning Optimal Lattice Vector Quantizers for End-to-end Neural Image Compression"
_NeurIPS.cc/2024/Conference — NeurIPS 2024 poster_

### Official Review · Reviewer_uEQ6 · 2024-07-11

**Soundness:** 2
**Presentation:** 2
**Contribution:** 2
**Rating:** 3
**Confidence:** 4

**Summary:**

This paper proposed to optimize the codebooks of lattice vector quantization (LVQ) for improved rate-distortion performance in neural image compression. Unlike previous LVQ methods using pre-designed codebook structure, the proposed method is able to adaptively learn the optimal codebook structure for nonuniform distribution. Experimental results demonstrate the proposed method outperforms existing quantization methods for neural image compression in terms of rate-distortion-complexity performance.

**Strengths:**

The idea of learnable codebook basis in lattice VQ is interesting. The proposed method has a significant performance gain and it is also compatible to various neural image codecs.

**Weaknesses:**

My first concern is about the correctness of entropy model design. According to Section 3.4, the distribution of the lattice vector m is factorized into the product of the distributions of its independent variables, which are modeled as a product of Gaussian Mixture Models.
The problem is how do you model the integer vector m as a continuous distribution? In previous methods, the quantized vector of uniform quantization is usually modeled as a Gaussian convolved with a uniform distribution, to match the integral process from continuous distribution of y to discrete distribution of y_hat. But in your paper, especially in Eq. (10, 12, 13), I don’t think it’s a typo caused by you forgetting the integral process. The rate is estimated by directly calculating the log density, which is far away from the practical rate. Moreover, in the ablation study (Table 2 and Table 4), how do you estimate the rate of traditional lattice VQ and general VQ under similar neural image coding system? Both the ablations can also achieve significant performance gain. Could you provide a description about the implementation details?

My second concern is about the effectiveness of orthogonal constraint. While I understand the orthogonal constraint can largely reduce coding complexity, the decorrelation ability should also be significantly weakened. According to Eq. (6), the quantization process is equivalent to: first rounding the linearly transformed latent and then inversely transform the rounding result. From a practical view, it's not much different from uniform scalar quantization + two MLP layers. How about the performance without orthogonal constraint?
Finally, it would be better to provide the rate-distortion curves.

**Questions:**

Please refer to weakness part.

**Limitations:**

This paper seems to lack discussions of the limitations or the potential negative societal impact.

---

> ### Author Rebuttal · Authors · 2024-08-07
>
> - **W1. [Omission of integral operation]**
> Thanks for pointing out the lack of rigor in our math notation.  Hope you can understand that it is only an innocent omission of an integral symbol.
> We will write out, in the final version, the corrected probability mass function (PMF) over the lattice quantized vector:
> $$ p_{m} = \int_{V(m)} \sum_{k=1}^{K} \mathbf{\Phi}_k \mathcal{N} (x; \mu_k, \Sigma_k) dx $$
>
>     $$ p_{m} = \prod_{i=1}^n \int_{V(m^i)} \sum_{k=1}^{K} \mathbf{\phi_k}^i \mathcal{N} (x^i; \mu_k^i, \sigma_k^i) dx^i $$
>     where $V(m^i) = \lbrace x^i | Q(x^i)=m^i \rbrace$.
> &nbsp;
> - **W2. [Rate estimation of traditional LVQ and GVQ]**
> The rate estimation for traditional lattice VQ follows the method described in LVQ-VAE [1]. For general VQ, the rate estimation method follows the approach outlined in NVTC [2]. For more implementation details, please refer to their respective papers.
> &nbsp;
> - **W3. [Effectiveness of orthogonal constrai]**
> Sorry, we should have made this point more clearly.
> **The orthogonal constraint is needed to reduce the gap between lattice vector quantization during training and inference.**  As stated in Section 3.3, because lattice vector quantization is not differentiable (and thus cannot be directly integrated into end-to-end training), we propose a differentiable alternative using Babai’s Rounding Technique (BRT). BRT estimates a vector that is sufficiently close to, but not necessarily exact, the closest lattice point. However, our experiments revealed that the BRT-estimated lattice vector may be far away from the exact closest lattice point if the cell shape of the optimized LVQ is arbitrary and unconstrained. This discrepancy causes an inconsistency between training and inference, if the learnt lattice vector quantizer is employed in the inference stage.
> As analyzed in [3], the BRT-estimated lattice point will match the closest lattice point if the basis vectors of the generator matrix are mutually orthogonal. For this reason, we propose to impose orthogonal constraints on the basis vectors of the generator matrix, enhancing the accuracy of the BRT-estimated lattice point and reducing the gap between lattice vector quantization during training and inference.
>  **The performance without the orthogonal constraint will drop by about 4.2\% for BD-rate.** We report the detailed performance numbers with and without the orthogonal constraint in the following table.
> We will make the above clarifications in the final version.
> &nbsp;
> | Entropy Model |  Bmshj2018 w/ orthogonal constraint |  Bmshj2018 w/o orthogonal constraint | SwinT-ChARM w/ orthogonal constraint |  SwinT-ChARM w/o orthogonal constraint |
> |  ----  | ----  | ----  |  ----  | ----  |
> | Factorized (w/o context) | -22.60% | -18.2% | -12.44% | -8.09% |
> | Checkerboard | -11.94% | -7.94% | -6.37% |  -2.25% |
> | Channel-wise Autoregressive | -10.61% | -6.34% | -5.61% | -1.22% |
> | Spatial-wise Autoregressive | -8.31% | -4.20% | -2.31% | +2.08% |
> &nbsp;
>
> - We provide the RD-curves of different methods in the rebuttal PDF. It can be seen that the proposed optimal lattice vector quantization consistently improves the existing compression models in both low and high bitrate cases. However, the performance gain is more pronounced at lower bitrates.
> &nbsp;
>
> [1]. Kudo, S., Bandoh, Y., Takamura, S., \& Kitahara, M. LVQ-VAE: End-to-end Hyperprior-based Variational Image Compression with Lattice Vector Quantization. OpenReview 2023.
> [2]. Feng, R., Guo, Z., Li, W., \& Chen, Z. NVTC: Nonlinear vector transform coding. CVPR 2023.
> [3]. Babai L. On Lovász’ lattice reduction and the nearest lattice point problem. Combinatorica, 1986.

---

> ### Comment · Area_Chair_mQdZ · 2024-08-13
>
> Dear Reviewer,
>
> The authors have posted an author response here. Could you go through the rebuttal and update your opinion and rating as soon as possible?
>
> Your AC

---

> ### Author Response · Authors · 2024-08-14
>
> Dear Reviewer,
>
> We hope that our rebuttal has effectively addressed your concerns. We are grateful for the time and effort you’ve dedicated to reviewing our paper. Your valuable feedback and suggestions have been very helpful, and we will certainly incorporate them into our future revisions. Thank you once again for your time.

---

> ### Comment · Reviewer_uEQ6 · 2024-08-14
> **thanks for the response**
>
> Thanks for your replying. I have carefully read all the review and rebuttal. I still have concern about the entropy model design and orthogonal constraint. Additionally, the rebuttal only provides RD curves for the bmshj2018 model. Could you also provide the RD curves for the SOTA model LIC-TCM mentioned in Table 1?
>
> Based on the revised PMF function and your response W2 to Reviewer B5jh, it seems that you factorized the distribution of quantized latent vectors instead of the distribution of integer indices during training. This implies that you approximate the shape of integration area from Voronoi polyhedron to hypercube. Such an approximation would lead to incorrect rate estimation results. Only the scalar quantization could perfectly match the hypercube integration area. LVQ with orthogonal matrix would rotate and scale the hypercube, not to mention LVQ with other matrices.
> Moreover, in rebuttal you claim that orthogonal constraint is used to reduce train-test mismatch of BRT-estimated lattice point and exact lattice point. The test-time LVQ would further increase the gap between the approximated PMF and actual PMF. I am curious to know how you calculate the integration over the Voronoi polyhedron area during inference.
>
> I would prefer to keep the score unless these concerns are fully addressed.

---

> > ### Author Response · Authors · 2024-08-14
> >
> > With only a few hours remaining in the discussion phase, we hope this additional clarifications can address your concerns. We are happy to answer any further questions if needed.

---

> ### Author Response · Authors · 2024-08-14
>
> Thanks for your feedback. We are glad to clarify your further concerns regarding the entropy model design and orthogonal constraint.
> &nbsp;
>
> - RD curves
> The RD curves for the SOTA model LIC-TCM follow a similar trend to the bmshj2018 model, although the gains are smaller at low bitrates. Unfortunately, we are unable to upload images or files at this stage, so we cannot provide the RD curves right now. However, we will include these RD curves and discuss them in future revisions.
> &nbsp;
>
> - Entropy model and orthogonal constraint
> You are correct in understanding that we factorized the distribution of quantized latent vectors. However, the integration area is not a simple hypercube; it’s more accurately described as a hyperrectangle, with different lengths for each dimension. We recognize that approximating the integral over such an area cannot precisely match the true region of LVQ. This approximation is a necessary compromise during training, as integrating over a dynamically changing region of arbitrary shape and size is impractical (at least to us). While this does introduce a gap between training and inference, the orthogonal constraint helps mitigate this by constraining the LVQ shape to a rotated hyperrectangle, though some discrepancy remains due to rate estimation over the non-rotated hyperrectangle.
> &nbsp;
> During inference, we calculate the integration over the arbitrarily shaped Voronoi polyhedron using Monte Carlo integration, a numerical integration method involving random sampling, as also employed in LVQ-VAE.  It is noteworthy that the rate estimation for traditional lattice VQ follows the same approach in our experiments.
> &nbsp;
>
>
> References
>
> [1]. Kudo, S., Bandoh, Y., Takamura, S., & Kitahara, M. LVQ-VAE: End-to-end Hyperprior-based Variational Image Compression with Lattice Vector Quantization. OpenReview 2023.

---

### Official Review · Reviewer_8Aj7 · 2024-07-12

**Soundness:** 3
**Presentation:** 3
**Contribution:** 3
**Rating:** 6
**Confidence:** 3

**Summary:**

- This paper proposes a method of differential lattice vector quantization for DNN image compression.
- The authors apply BRT [5] to estimate a vector in the basis vectors B that is orthogonal, close to vector v.
- Experiments in Table 1 compare the proposed method with scalar quantizers, and experiments in Table 2 compare the proposed method with classical pre-defined lattice vector quantizers, and claim that the proposed method is valid for BD-rate.
- The authors claim that they have confirmed the effectiveness of the proposed method in BD-rate.

**Strengths:**

- Quantization is one of the fundamental and important methods in DNN image compression and this paper attempts to introduce the Lattice Vector Quantizer to DNN image compression, which is challenging and interesting.

**Weaknesses:**

- The evaluation of Table1 and Table2 is based on the Kodak and CLIC datasets, which are commonly used in this field, but these datasets may should be evaluated separately, as is often done in other papers for comparison.
- It is unfortunate that the experimental results showing the effect of quantization are presented only for the final BD-rate. If only BD-rate was used, the evaluation conditions would not be clear, such as the range of bitrates evaluated. For example, it would have been desirable to show comparisons with other methods using RD-curve.
- Furthermore, it would be more convincing if an analysis of the characteristics and behavior of the quantized values from experiment was conducted.

**Questions:**

- Why does  imposing orthogonal constraints on the basis vectors lead to enhance the accuracy and improve training stability?

**Limitations:**

- Section 4.5 discusses the limitations and says that the effect is modest for heavy-duty compression models, but it would be better to show what a heavy-duty model is if it can be shown in more detail.

---

> ### Author Rebuttal · Authors · 2024-08-07
>
> - **W1. [Separated evaluation]**
> We provide the results which are separately evaluated in the following table (BD-rate over uniform scalar quantizer). It can be observed that the performance trend is consistent across the separate and combined evaluations.
> &nbsp;
> | Entropy Model |  Bmshj2018 Kodak |  Bmshj2018 CLIC | SwinT-ChARM Kodak |  SwinT-ChARM CLIC |
> |  ----  | ----  | ----  |  ----  | ----  |
> | Factorized (w/o context) | -20.61% | -23.15% | -11.16% | -13.12% |
> | Checkerboard | -10.53% | -12.49% | -5.28% |  -6.74% |
> | Channel-wise Autoregressive | -10.32% | -11.21% | -4.33% | -6.12% |
> | Spatial-wise Autoregressive | -7.12% | -8.97% | -1.98% | -2.65% |
> &nbsp;
> - **W2. [R-D curves]**
> We provide the RD-curves of different methods in the rebuttal PDF. It can be seen that the proposed optimal lattice vector quantization consistently improves the existing compression models in both low and high bitrate cases. However, the performance gain is more pronounced at lower bitrates.
> &nbsp;
> - **W3. [Characteristics of generator matrix]**
> Thanks for your great suggestion. We analyze the deformation of the learned generator matrix by measuring the ratio of the longest to the shortest basis vectors, which turned out to be 6.8, indicating extreme asymmetry in the optimized lattice structure. Additionally, the angle between the longest and the shortest basis vectors is approximately 60 degrees. These characteristics of the learned generator matrix indicate that the cell shape of the optimized LVQ is significantly different from that of a uniform scalar quantizer.
> &nbsp;
> - **Q1. [Effectiveness of orthogonal constraint]**
> Sorry, we should have made this point more clearly.
> **The orthogonal constraint is needed to reduce the gap between lattice vector quantization during training and inference.**  As stated in Section 3.3, because lattice vector quantization is not differentiable (and thus cannot be directly integrated into end-to-end training), we propose a differentiable alternative using Babai’s Rounding Technique (BRT). BRT estimates a vector that is sufficiently close to, but not necessarily exact, the closest lattice point. However, our experiments revealed that the BRT-estimated lattice vector may be far away from the exact closest lattice point if the cell shape of the optimized LVQ is arbitrary and unconstrained. This discrepancy causes an inconsistency between training and inference, if the learnt lattice vector quantizer is employed in the inference stage.
> As analyzed in [1], the BRT-estimated lattice point will match the closest lattice point if the basis vectors of the generator matrix are mutually orthogonal. For this reason, we propose to impose orthogonal constraints on the basis vectors of the generator matrix, enhancing the accuracy of the BRT-estimated lattice point and reducing the gap between lattice vector quantization during training and inference.
>  **The performance without the orthogonal constraint will drop by about 4.2\% for BD-rate.** We report the detailed performance numbers (BD-rate over uniform scalar quantizer) with and without the orthogonal constraint in the following table.
> We will make the above clarifications in the final version.
> &nbsp;
> | Entropy Model |  Bmshj2018 w/ orthogonal constraint |  Bmshj2018 w/o orthogonal constraint | SwinT-ChARM w/ orthogonal constraint |  SwinT-ChARM w/o orthogonal constraint |
> |  ----  | ----  | ----  |  ----  | ----  |
> | Factorized (w/o context) | -22.60% | -18.2% | -12.44% | -8.09% |
> | Checkerboard | -11.94% | -7.94% | -6.37% |  -2.25% |
> | Channel-wise Autoregressive | -10.61% | -6.34% | -5.61% | -1.22% |
> | Spatial-wise Autoregressive | -8.31% | -4.20% | -2.31% | +2.08% |
> &nbsp;
>
> [1]. Babai L. On Lovász’ lattice reduction and the nearest lattice point problem. Combinatorica, 1986.

---

> > ### Comment · Reviewer_8Aj7 · 2024-08-11
> > **Thank you very much for your thoughtful response.**
> >
> > Many of my original concerns have been addressed.
> > However, I am also concerned about the entropy estimation model that reviewer uEQ6 pointed out.
> > Are the results in Tables 1, 2, etc. the result of actual encoding using arithmetic coding [40, 44], rather than theoretical values ​​based on the output of an entropy model? And have you verified that these encoded data can be decoded correctly?

---

> > > ### Author Response · Authors · 2024-08-11
> > >
> > > Thank you for your feedback. The results presented in Tables 1 and 2 were obtained using arithmetic coding (specifically, range coding), rather than being theoretical values estimated from the output of the entropy model. Additionally, we have verified that the arithmetic encoded data can be decoded correctly.

---

> > > > ### Comment · Reviewer_8Aj7 · 2024-08-11
> > > >
> > > > Thank you for your response. I have updated my rates.

---

> > > > > ### Author Response · Authors · 2024-08-14
> > > > >
> > > > > We are pleased that your concerns have been addressed and will incorporate your feedback and suggestions in future revisions. Your insights have significantly helped improve our paper in many aspects. We sincerely appreciate your decision to update your rate. Thank you!

---

### Official Review · Reviewer_4Wxt · 2024-07-12

**Soundness:** 3
**Presentation:** 3
**Contribution:** 2
**Rating:** 5
**Confidence:** 3

**Summary:**

In this paper, the use of lattice vector quantization (LVQ) with deep learning-based image compression has been studied. The authors proposed to learn the bases of the LVQ matrix, for changing the quantization cells to better capture feature correlations and thus better coding. The experiments demonstrated significant average improvement for applying the proposed method to existing deep learning based compression models with different configurations.

**Strengths:**

- Although vector quantization could potentially be a better solution than uniform scalar quantization, its use has not been well-studied in the neural compression literature. The method proposed in this paper is simple but efficient for improving the rate-distortion (RD) performance in image compression. It only requires simple modifications to existing models with scalar quantization.
- The proposed method can be applied to different networks with various context models. The experiments also demonstrate performance gains in all configurations.

**Weaknesses:**

- The proposed method has not been directly compared with other vector quantization method for image compression (except for the general vector quantization).
- The performance of the proposed model at different bitrates is not provided. It is not clear whether the proposed method can consistently improve the compression model in both low and high bitrate cases.
- The impact of $\lambda_2$ and the term $L$ in the loss function has not been studied. How do they impact the performance?

**Questions:**

- In Table 3, how were the inference times measured? Are the GPUs fully utilized? 5ms looks too long for a GPU to perform only uniform scalar quantization.
- Are the BD-rates measured in PSNR?

**Limitations:**

The authors discussed the limitations and potential impact. There are also no potential negative societal impact.

Overall, I think the paper provides a simple but effective method and demonstrates promising results (although I am not especially familiar with the topic). The main issue is the lack of detailed rate-distortion performance analysis and the study of hyper-parameters. Additionally, it would be beneficial to discuss any potential impact on qualitative performance, as in image compression, quantitative results are not the sole criterion.

---

> ### Author Rebuttal · Authors · 2024-08-07
>
> - **W1. [Compared with other VQ based image compression methods]**
> A good suggestion. We have addressed your concern by adding new comparison results with the following vector quantization-based image compression methods: SHVQ (NeurIPS ’17) [1] and McQUIC (CVPR ’22) [2]. From the provided table (BD-rate over uniform scalar quantizer), it can be observed that the proposed learnable LVQ method not only outperforms these VQ-based image compression methods in terms of BD-rate but also demonstrates superior performance in computational complexity. As shown, our method achieves lower BD-rates, i.e., better compression performance. At the same time, the computational complexity of our approach is significantly lower than SHVQ and McQUIC, making it more practical, especially on resource-constrained devices.
> &nbsp;
> | Quantizers |  Bmshj2018 Factorized  |  Bmshj2018 Checkerboard  | SwinT-ChARM Factorized |  SwinT-ChARM Checkerboard |  Inference time |
> |  ----  | ----  | ----  |  ----  | ----  |  ----  |
> | Optimized LVQ | -20.49% | -10.71% | -10.68% | -5.89% | ~40ms |
> | SHVQ [1] | -8.23% | -5.18% | -6.55% | -1.92% | ~328ms |
> | McQUIC [2] | -15.11% | -7.93% | -7.24% | -3.18% |  ~771ms |
> &nbsp;
> - **W2. [Performance at different bitrates]**
> To address your concern, we provide the rate-distortion curves of different quantization methods in the rebuttal PDF. It can be observed that the proposed optimal lattice vector quantization consistently improves the existing compression models in both low and high bitrate cases. However, the performance gain is more pronounced at lower bitrates.
> &nbsp;
> - **W3. [Impact of $\lambda_2$ and the loss term L]**
> To address this, we have conducted ablation studies to thoroughly examine how these two hyperparameters affect the performance of our method.
>   - For $\lambda_2$, we employed a search strategy to identify the optimal value that yields the best performance. Our findings indicate that both excessively large and excessively small values of $\lambda_2$ can lead to performance degradation. When $\lambda_2$ is too large, it overly constrains the shape of the optimized lattice codebook, resulting in a sub-optimal lattice structure. Conversely, when $\lambda_2$ is too small, it loosens the orthogonal constraint, causing a significant gap between lattice vector quantization during training and inference (Please refer to the global rebuttal for more details).
>   - Regarding the loss term L, our initial experiments utilized the MSE metric. To further investigate, we have now included a study using the SSIM metric as the loss term L. The results from this study indicate that our conclusions drawn from using the MSE metric remain valid when using the SSIM metric. Specifically, the performance trends and the effectiveness of our proposed method are consistent across both metrics (Please refer to the rebuttal PDF for the R-D curves of SSIM metric).
> &nbsp;
> - **Q1. [Details of inference time]**
> Sorry for missing the details. The quantization times are not measured for quantizing a single token; instead, they are measured for quantizing a $512 \times 512 \times 1$ feature map. The GPU utilization is approximately 25\%, 28\%, and 42\% for Uniform Scalar Quantization (USQ), Lattice Vector Quantization (LVQ), and General Vector Quantization (GVQ), respectively.
> &nbsp;
> - **Q2. [Metric for BD-rate]**
>  Yes, the BD-rates are measured in PSNR.
> &nbsp;
>
> [1]. Agustsson, E., Mentzer, F., Tschannen, M., et al. Soft-to-hard vector quantization for end-to-end learning compressible representations. NeurIPS 2017.
> [2]. Zhu, X., Song, J., Gao, L., et al. Unified multivariate gaussian mixture for efficient neural image compression. CVPR 2022.

---

> > ### Comment · Reviewer_4Wxt · 2024-08-12
> >
> > Thank you for your reply. Here are some further questions:
> >
> > W1: I noticed that the reported RD here differs from the one in Table 1 of the paper (e.g., -22.60% vs. -20.49%). Is this difference due to the randomness of the experiments? Do the experiments exhibit a high degree of variability?
> >
> > W2: The provided RD curves only include results from the Bmshj2018 model. While this model shows the greatest gain from the proposed LVQ method (as shown in Table 1 of the paper), its improvement is relatively small at high bitrates (and sometimes there is no gain from USQ). Is the performance gain still consistent at high bitrates for stronger models like SwinT-ChARM or LIC-TCM? Additionally, please consider including RD curves for these models in future versions of the paper, as improvements on the latest models would be insightful.
> >
> > Q1: Why not perform quantization for all channels at once? And doesn’t LVQ perform quantization in groups of channels? Given the computational power of modern GPUs, I am not sure whether this is a reasonable setting. Please consider refining the testing methodology.

---

> > > ### Author Response · Authors · 2024-08-12
> > >
> > > Thank you for your additional questions. We appreciate the opportunity to clarify these points.
> > >
> > > W1: The difference in RD values arises from the use of different LVQ dimensions in our experiments. Specifically, the results in Table 1 of the paper are based on 32-dimensional LVQ, whereas the table in the rebuttal reflects results from 24-dimensional LVQ. Regarding the potential variability in the experiments, we conducted multiple trials for each setting, and the observed variability is generally less than 2% on average. In future versions of the paper, we will ensure consistency in experimental settings, such as the LVQ dimensions, to minimize any potential confusion. Additionally, we will include a brief discussion on the variability to further clarify this point.
> > >
> > > W2: We acknowledge that the performance improvement at high bitrates for the Bmshj2018 model is relatively small. Theoretically, uniform scalar quantization (USQ) can approach rate-distortion optimality at very high bitrates, meaning the advantages of vector quantization (including LVQ) over USQ tend to diminish in this regime. Therefore, the observed results align with these theoretical expectations.
> > > For stronger models such as SwinT-ChARM and LIC-TCM, the performance gains from LVQ remain consistent even at high bitrates. However, similar to the Bmshj2018 model, the gain decreases and may become negligible when the rate exceeds 1 bpp. We agree with your suggestion and will include RD curves for these stronger models in future versions of the paper to provide a more comprehensive comparison.
> > >
> > > Q1: LVQ can perform quantization in groups of either channels or adjacent spatial pixels. In our experiments, we evaluated both approaches and found that quantizing adjacent spatial pixels tends to offer slightly better performance than quantizing groups of channels. However, due to the complexity involved, we limited the quantization dimensions to 8, 16, 24, and 32 in our experiments. Given that the latent feature map typically has a dimension of 128 or higher, performing quantization across all channels simultaneously would introduce significant computational complexity and slow down LVQ, making it less practical. We will clarify this in future versions of the paper and will explore ways to leverage the computational power of modern GPUs to accelerate higher-dimensional LVQ.
> > >
> > > We hope these clarifications address your concerns, and we are happy to provide any additional information if needed.

---

> ### Author Response · Authors · 2024-08-14
>
> Dear Reviewer,
>
> We hope the additional clarifications appropriately addressed your further questions.
> We appreciate the time and effort you’ve dedicated to reviewing our paper. Your valuable feedback and suggestions have been immensely helpful, and we will certainly incorporate them into our future revisions. Thank you again for your time.

---

### Official Review · Reviewer_B5jh · 2024-07-22

**Soundness:** 3
**Presentation:** 3
**Contribution:** 3
**Rating:** 6
**Confidence:** 4

**Summary:**

In this paper, the authors proposed learning the lattice vector quantization for the end-to-end neural image compression. Their method is based on the learning the orthogonal basis function, and each vector is represented as the linear combination of the basis function during training, and they used orthogonality of the basis function to decompose the marginal of the entropy distribution. They showed that their method outperformed existing non-learned lattice vector quantization, and their proposal has significant gain in the simple entropy model (Factorized entropy model) but for the complex autoregressive entropy model their gains are less.

**Strengths:**

1) The paper proposed learning based lattice vector quantization for the end-to-end neural image compression. The code books of the lattice vector quantization are assumed to be orthogonal basis functions, and the use of the Babai’s Rounding Technique for the quantization is interesting during the training time.
2) Marginalization of the multi-variate gaussian distribution entropy model due to the orthogonality of the basis functions
2) Bit-rate savings over the pre-defined lattice vector quantization

**Weaknesses:**

1) The performance of the lattice vector quantization is compared with scalar quantization, non-learned lattice vector quantization, and general vector quantization. The description of the general vector quantization or reference to it is missing in the experimental section, to know which paper is used comparison.
2) The authors mentioned that they learn the entropy model over the quantized latent vectors with learned lattice vector quantization, but in the experiments, authors mentioned that they code the integer combination of indices in the bitstream. The learned entropy model for the quantized latent vectors might not be optimal to code the integer combination of indices.
3) The performance of the proposed method does not have significant gain in the context models, as the most recent end-to-end neural image compression methods are based on the autoregressive model, thus applying the proposed method might not have a significant gain.
4) The toy example plots are missing in the paper, which would help to understand the structure of the partition generated by the proposed method.

**Questions:**

1) The proposed method has significant gain with respect to the factorized entropy model compared to the autoregressive entropy model, so whether the proposed method on the factorized entropy model can approximate the performance of the autoregressive entropy model? what will be the performance gap b/w the proposed method on the factorized entropy model and the autoregressive entropy model? if the performance gap is small, then whether the proposed method can be used to replace the autoregressive entropy model?

2) Which method or paper did you use to compute the results of the generalized vector quantization ?

**Limitations:**

the limitations are addressed

---

> ### Author Rebuttal · Authors · 2024-08-07
>
> - **W1. [General vector quantization]**
> We build the general vector quantization model based on NVTC (CVPR'23) [1].
> &nbsp;
> - **W2. [Entropy model]**
> Sorry for the discrepancy. The entropy model is optimized over the lattice quantized vectors and used to code the same vectors into the bitstream. The idea of coding the integer combination coefficients is still under investigation.
> &nbsp;
> - **W3. [Limited gain with autoregressive model]**
> We acknowledge this limitation and have discussed it in the paper.
> However, as well known, context models, especially autoregressive ones, tend to slow down the coding speed dramatically. Our method is meant to build LIGHTWEIGHT models for deployment on edge devices. As these devices have severe resource constraints, the model efficiency and simplicity become paramount. Forgoing computationally intensive context models offers a practical solution for real-time applications on such devices.
> &nbsp;
> - **W4. [Toy example plots]**
>  We appreciate the value of visualizing the lattice structure generated by the proposed method. However, plotting the lattice structures in more than three dimensions remains a challenge; in fact, we have not seen such drawings in any prior literature on the topic.
> &nbsp;
> - **Q1. [Optimized LVQ + factorized model vs. autoregressive model]**
> We have included R-D curves of different methods in the rebuttal PDF to address your concerns. It can be observed that the proposed learnable LVQ method coupled with the factorized entropy model can approach the performance of the autoregressive entropy model, particularly at low bitrates. For high bitrates, the autoregressive entropy model still has an edge.
> Given this, we believe that the proposed LVQ method can serve as a viable alternative to the autoregressive entropy model, especially in scenarios where low bitrate performance is crucial and computational efficiency is a priority. The factorized entropy model, being less computationally intensive, combined with our LVQ method, offers an attractive trade-off between performance and efficiency.
> &nbsp;
>
> [1]. Feng, R., Guo, Z., Li, W., \& Chen, Z. NVTC: Nonlinear vector transform coding. CVPR 2023.

---

> ### Comment · Area_Chair_mQdZ · 2024-08-13
>
> Dear Reviewer,
>
> The authors have posted an author response here. Could you go through the rebuttal and update your opinion and rating as soon as possible?
>
> Your AC

---

> ### Author Response · Authors · 2024-08-14
>
> Dear Reviewer,
>
> We hope the rebuttal has appropriately addressed your concerns. Thank you for the time and effort you have invested in reviewing our paper. Your insightful feedback and suggestions have been extremely beneficial, and we will make sure to include them in our future revisions. We appreciate your continued support and time.

---

### Author Rebuttal · Authors · 2024-08-07

- We would like to thank the reviewers for their valuable feedback, and are encouraged by their positive reception of our work.
&nbsp;

- We respond below to specific points raised by each reviewer. We hope we addressed all the reviewers' concerns, and we will be happy to provide additional clarifications upon request.
&nbsp;

- We provide R-D curves of different methods in the PDF file, as requested by multiple reviewers.  It can be observed from these R-D curves that the proposed optimal lattice vector quantization consistently improves the existing compression models in both low and high bitrate cases. However, the performance gain is more pronounced at lower bitrates.  It can also be observed that the proposed learnable LVQ method coupled with the factorized entropy model can approach the performance of the autoregressive entropy model, particularly at low bitrates. For high bitrates, the autoregressive entropy model still has an edge.
Given this, we believe that the proposed LVQ method can serve as a viable alternative to the autoregressive entropy model, especially in scenarios where low bitrate performance is crucial and computational efficiency is a priority. The factorized entropy model, being less computationally intensive, combined with our LVQ method, offers an attractive trade-off between performance and efficiency.
&nbsp;

- We conduct ablation studies to thoroughly examine how two hyperparameters ($\lambda_2$ and the loss term L) affect the performance of our method.
  - For $\lambda_2$, we employed a search strategy to identify the optimal value that yields the best performance. Our findings indicate that both excessively large and excessively small values of $\lambda_2$ can lead to performance degradation. When $\lambda_2$ is too large, it overly constrains the shape of the optimized lattice codebook, resulting in a sub-optimal lattice structure. Conversely, when $\lambda_2$ is too small, it loosens the orthogonal constraint, causing a significant gap between lattice vector quantization during training and inference (Please refer to the global rebuttal for more details).
  - Regarding the loss term L, our initial experiments utilized the MSE metric. To further investigate, we have now included a study using the SSIM metric as the loss term L. The results from this study indicate that our conclusions drawn from using the MSE metric remain valid when using the SSIM metric. Specifically, the performance trends and the effectiveness of our proposed method are consistent across both metrics (Please refer to the rebuttal PDF for the R-D curves of SSIM metric).
&nbsp;

- Here we make a clarification about the effectiveness of the proposed orthogonal constraint, which are requested by multiple reviewers.
**The orthogonal constraint is needed to reduce the gap between lattice vector quantization during training and inference.**  As stated in Section 3.3, because lattice vector quantization is not differentiable (and thus cannot be directly integrated into end-to-end training), we propose a differentiable alternative using Babai’s Rounding Technique (BRT). BRT estimates a vector that is sufficiently close to, but not necessarily exact, the closest lattice point. However, our experiments revealed that the BRT-estimated lattice vector may be far away from the exact closest lattice point if the cell shape of the optimized LVQ is arbitrary and unconstrained. This discrepancy causes an inconsistency between training and inference, if the learnt lattice vector quantizer is employed in the inference stage.
As analyzed in [1], the BRT-estimated lattice point will match the closest lattice point if the basis vectors of the generator matrix are mutually orthogonal. For this reason, we propose to impose orthogonal constraints on the basis vectors of the generator matrix, enhancing the accuracy of the BRT-estimated lattice point and reducing the gap between lattice vector quantization during training and inference.
 **The performance without the orthogonal constraint will drop by about 4.2\% for BD-rate.** The detailed performance numbers (BD-rate over uniform scalar quantizer) with and without the orthogonal constraint are tabulated below.
We will make the above clarifications in the final version.

[1]. Babai L. On Lovász’ lattice reduction and the nearest lattice point problem. Combinatorica, 1986.

---

### Author Response · Authors · 2024-08-14
**Acknowledgment and Gratitude to all Reviewers and AC**

Dear Reviewers and AC,

As the discussion phase draws to a close, we want to express our sincere gratitude for your time and effort in reviewing our paper. Your valuable feedback and suggestions have been incredibly helpful, and we will certainly incorporate them into our future revisions. Thank you once again for your thoughtful insights.

---

### Decision · Program_Chairs · 2024-09-25

**Decision:**

Accept (poster)

**Comment:**

The paper presents a novel approach to lattice vector quantization (LVQ) for neural image compression, demonstrating significant improvements in rate-distortion performance. The use of orthogonal basis functions and Babai's Rounding Technique during training is particularly noted for its innovation. The learned LVQ performs better than non-learned lattice vector quantizers, as indicated in the experiments. The paper receives two weak accept, one borderline accept, and one reject. The AC examines the comments and believes the major concerns should have been addressed. Thus, the AC recommends the acceptance of the paper.